

# Unpredictable soil conditions can affect the prevalence of a microbial symbiosis

Trey J. Scott[1,2], Calum J. Stephenson[2], Sandeep Rao[2], David C. Queller[2] and Joan E. Strassmann[2]

[1] Department of Organismic and Evolutionary Biology, Harvard University, Cambridge, MA, United States
[2] Department of Biology, Washington University in St. Louis, St. Louis, MO, United States

## ABSTRACT

The evolution of symbiotic interactions may be affected by unpredictable conditions. However, a link between prevalence of these conditions and symbiosis has not been widely demonstrated. We test for these associations using *Dictyostelium discoideum* social amoebae and their bacterial endosymbionts. *D. discoideum* commonly hosts endosymbiotic bacteria from three taxa: *Paraburkholderia, Amoebophilus* and Chlamydiae. Three species of facultative *Paraburkholderia* endosymbionts are the best studied and give hosts the ability to carry prey bacteria through the dispersal stage to new environments. *Amoebophilus* and Chlamydiae are obligate endosymbiont lineages with no measurable impact on host fitness. We tested whether the frequency of both single infections and coinfections of these symbionts were associated with the unpredictability of their soil environments by using symbiont presence-absence data from *D. discoideum* isolates from 21 locations across the eastern United States. We found that symbiosis across all infection types, symbiosis with *Amoebophilus* and Chlamydiae obligate endosymbionts, and symbiosis involving coinfections were not associated with any of our measures. However, unpredictable precipitation was associated with symbiosis in two species of *Paraburkholderia*, suggesting a link between unpredictable conditions and symbiosis.

## INTRODUCTION

The evolution of cooperation varies with ecological unpredictability (*Scott, 2023*; *Scott et al., 2023b*). For example, the prevalence of cooperative breeding in birds is associated with unpredictable environmental conditions (*Jetz & Rubenstein, 2011*; *Griesser et al., 2017*). Cooperative breeding is thought to allow organisms to invade unpredictable environments (*Cornwallis et al., 2017*) or buffer against times when conditions are harsh (*Capilla-Lasheras et al., 2021*). So far studies on the relationship between ecological unpredictability and cooperation have focused on interactions between members of the same species (*Jetz & Rubenstein, 2011*; *Sheehan et al., 2015*; *Griesser et al., 2017*; *Firman et al., 2020*). Cooperative associations between different species in a symbiosis, or mutualism, has been suggested to have similar benefits in unpredictable environments

Corresponding author
Trey J. Scott, tjscott@wustl.edu

(*Lekberg & Koide, 2014*; *Veresoglou et al., 2021*; *Scott, Queller & Strassmann, 2022a*) and may thus be associated with them. However, this association has not been tested.

We investigated whether symbiosis was associated with unpredictable conditions using the microbiome of *Dictyostelium discoideum*. *D. discoideum* is a social amoeba that spends part of its lifecycle as a single cell eating bacteria in the soil (*Raper, 1937*). After exhausting edible bacteria, individual amoebae come together and form a multicellular structure called a fruiting body to disperse resistant spores (*Kessin, 2001*). Inside some fruiting bodies in the wild, different species of bacteria have been identified (*Brock et al., 2018*; *Sallinger, Robeson & Haselkorn, 2021*; *Steele et al., 2023*). Most of these bacteria appear to be regular soil bacteria that happen to be in the matrix of the spore-containing part of the fruiting body. Many of these bacteria are even edible by *D. discoideum* (*Brock et al., 2018*).

A subset of the bacteria that are found in *D. discoideum* fruiting bodies appear to be prevalent symbionts. The first symbionts to be discovered were three species of facultatively endosymbiotic *Paraburkholderia* bacteria (*Brock et al., 2011*; *DiSalvo et al., 2015*). The life histories of these *Paraburkholderia* bacteria in their natural soil habitats are unknown, but they can be cultured outside of their hosts in the lab (*DiSalvo et al., 2015*; *Brock et al., 2020*) and one species has been shown to be horizontally transferred in the lab (*Noh et al., 2024*). Two of these *Paraburkholderia* species, *P. hayleyella* and *P. bonniea*, may have a longer history of host association as shown by their reduced genomes, while *P. agricolaris* may be a newer symbiont (*Noh et al., 2022*).

All three *Paraburkholderia* species increase host fitness by allowing hosts to carry other species of edible bacteria along with *Paraburkholderia* inside the spore-containing part of the fruiting body called a sorus (*Khojandi et al., 2019*; *Brock et al., 2020*). Paraburkholderia are often carried inside spores while prey bacteria are carried outside of the spores in the sorus (*Khojandi et al., 2019*). Carriage allows host amoebae to seed out populations of prey bacteria that hosts can then eat (*Brock et al., 2011*). However, the ability to carry comes at the cost of reduced spore production when edible bacteria are common (*DiSalvo et al., 2015*; *Scott, Queller & Strassmann, 2022b*). The source of this fitness cost for hosts is unknown, though there is some evidence that *Paraburkholderia* itself harms hosts. For example, the density of *Paraburkholderia* tends to be associated with lower host spore production (*Scott, Queller & Strassmann, 2022a*, *2022b*; *Noh et al., 2024*) and *Paraburkholderia* infection interferes with host immune cells that develop during the multicellular stage (*Scott et al., 2023a*).

*D. discoideum* also harbors endosymbiotic bacteria that are obligate: one from the genus *Amoebophilus* and different haplotypes from the phylum Chlamydiae. These obligate endosymbionts cannot be cultured outside of their hosts. Both *Amoebophilus* and Chlamydiae have not been found to measurably affect host fitness, even when they occur as coinfections with *Paraburkholderia* (*Haselkorn et al., 2021*). We will refer to these obligate endosymbionts as *Amoebophilus* and Chlamydiae.

Environmental sampling has found that *Paraburkholderia* prevalence is about 25% of sampled hosts but varies by sampling location (*Haselkorn et al., 2019*) and over time (*DuBose et al., 2022*). Obligate endosymbionts are found in about 40% of sampled hosts

(*Haselkorn et al., 2021*). *Paraburkholderia* and *Amoebophilus* coinfections are more common than expected due to chance (*Haselkorn et al., 2021*).

A key source of unpredictability in the soil environment of *D. discoideum* that has not been studied is precipitation. Precipitation can drastically shift the soil environment because of the complex structure and physical properties of the soil (*Or et al., 2007*). Such shifts are known to affect the abundance of microbes in the soil (*Zeglin et al., 2013*). When precipitation is unpredictable, it is likely to impact the availability of soil bacteria for *D. discoideum* to eat. We hypothesize that hosts that have *Paraburkholderia* symbionts may be buffered from unpredictable changes in prey abundance because they can carry prey bacteria (*Scott, Queller & Strassmann, 2022a*).

Other soil characteristics may also be important for the prevalence of symbiosis. pH has already been shown to affect the *D. discoideum* microbiome (*Sallinger, Robeson & Haselkorn, 2021*). Temperature can have strong effects on host amoebae that could shape their interactions with symbionts (*Shu et al., 2020*). Soil nutrients (usually measured by the ratio of carbon to nitrogen in the soil) have not been studied in this symbiosis but are known to affect other symbiotic interactions (*Johnson et al., 2010*) and to affect soil bacteria (*Bahram et al., 2018*) that are potential prey of *D. discoideum*. Here we test how unpredictability and other soil characteristics affect the symbiosis between *D. discoideum* and its symbionts.

## MATERIALS AND METHODS

### Presence of text from preprinted thesis chapter

Portions of the text in this manuscript were previously published as a preprint (*Scott et al., 2023b*) and as part of a thesis (*Scott, 2023*).

### Data acquisition and processing

To measure the frequency of symbiosis, we used data from prior environmental sampling (*Haselkorn et al., 2019*, *2021*). The first study (*Haselkorn et al., 2019*) tested *D. discoideum* isolates from 21 locations in the United States (one location was sampled two separate times) for the presence of the three species of *Paraburkholderia* symbionts (*Brock et al., 2020*) using *Paraburkholderia* specific 16S rRNA sequencing. The second study (*Haselkorn et al., 2021*) tested a similar set of *D. discoideum* isolates for *Amoebophilus* and Chlamydiae using symbiont specific 16S rRNA sequencing, but also included samples from a few additional countries. For this study, we focused only on the United States samples because sites from other countries were not well sampled and could skew the results. We used these data to construct a presence (1)-absence (0) variable for each *D. discoideum* clones for whether they were infected with any of the three species of *Paraburkholderia*, or *Amoebophilus*, or Chlamydiae. We also generated a presence-absence measure for total symbiosis (having any symbiont across all the tested taxa) and for coinfections between the five different symbiont types (*P. agricolaris*, *P. hayleyella*, *P. bonniea*, *Amoebophilus*, and Chlamydiae).

To investigate the role of environmental predictability on the *Dictyostelium-Paraburkholderia* symbiosis, we acquired data on long-term precipitation. We also

acquired data on soil pH, soil organic carbon, nitrogen, and temperature for each sample location from online databases (detailed below). These variables are known to affect the abundance of bacteria in the soil (*Bahram et al., 2018*). For each location, we collected monthly precipitation data from 1901 to 2020 from the climate research unit database version 4.05 (*Harris et al., 2020*). To measure the predictability of precipitation across these monthly measures, we calculated Colwell's P (*Colwell, 1974*) using the *Colwell's* function in the hydrostats package (*Bond & Bond, 2022*) with 12 bins corresponding to months and with log-transformed precipitation measures as in Table 2 in *Colwell (1974)*. Colwell's P ranges from completely unpredictable (0) to completely predictable (1). To better capture ecologically relevant timescales, we tested two P measures meant to capture long-term and recent predictability: (1) calculated with precipitation data from 1901 to the year that a sample was collected and (2) calculated from precipitation data from 5 years before the sample was taken.

We collected mean soil pH, mean nitrogen, and mean organic carbon data from the SoilGrids database version 2.0 (*de Sousa et al., 2020*). SoilGrids are soil predictions based on empirical soil measurements and are generated at 250-m scales. We collected mean soil temperature variables from *Lembrechts et al. (2022)* that were generated by predicting deviations of soil temperatures from air temperatures at 0 to 5 cm and 5–15 cm depths. We used 0–5 cm depths for soilGrids and soil temperature data because *D. discoideum* typically resides in the top layers of soil.

## Statistical methods

To test for coinfections across locations, we used mixed effect logistic regression from the *lme4* package (*Bates, 2010*) in R version 4.1.2 (*R Core Team, 2013*). We tested for possible coinfections between all five of the symbiont types that we investigated. To account for multiple observations at a location, we used location as a random effect. We treated the location that was sampled twice (Mountain Lake Biological Station) as two separate locations because soil samples were taken from different areas within Mountain Lake Biological Station and because samples were collected 14 years apart.

As a follow up to our logistic regression results across locations, we tested whether coinfections involving different *Paraburkholderia* species were random in specific locations using Fisher's exact tests (Table SI). To perform Fisher's exact tests, we constructed a 2 × 2 contingency table for each sampling location in which at least two of the investigated three *Paraburkholderia* symbionts were present. To correct for multiple comparisons, we adjusted *p*-values using Benjamini-Hochberg's correction.

To test for associations between soil characteristics and prevalence of individual symbionts or coinfections, we fit a set of mixed effect logistic regression models using lme4 (*Bates, 2010*) as above. In these models, we tested the effect of soil characteristics on each of our five symbiont types individually. We also tested the effect of soil characteristics on the prevalence of symbiosis regardless of the type and on the prevalence of coinfections between *P. hayleyella* and *Amoebophilus* as these coinfections were more common than expected by chance. We tested models that were derived from a full model that included the precipitation predictability (Collwell's P) since 1901 and for a 5 year period before
samples were collected, mean annual temperature (MAT), carbon to nitrogen ratio (C/N), mean annual precipitation (MAP), and soil pH. To reduce the risk of overfitting, we only compare models with two or fewer total predictors. To identify top models among the set derived from the full model, we used AICc values (*Burnham & Anderson, 2004*) and examined effect sizes of model estimates. We identify uninformative models if the model does not differ from an intercept only (null) model in terms of AICc. We identify informative models if the model AICc is less than the null model by two or more. For multiple models that fit better than the null model, we examined models within two AICc units of the best fitting model and looked for variables that were consistently in top models. To test for spatial autocorrelation in our models, we performed a Moran's I test on simulated residuals using the *DHARMa* package in R (*Hartig, 2020*). All models were free of spatial autocorrelation.

For models that showed an effect of unpredictable precipitation on *P. hayleyella* and *P. agricolaris* prevalence, we ensured that these effects were not solely due to the influence of the two largest sample locations (Mountain Lake Biological Station in Virginia). To do this, we refit 1,000 models on subsets of the data where Mountain Lake samples were no longer outliers in terms of the number of sampled clones. To produce subsets, we randomly removed 350 clones from the pool of clones from both locations. Using our fit models, we then estimated the effect of unpredictable precipitation and compared these estimated effects to those estimated from the full dataset.

To test whether *P. hayleyella* and *P. bonniea* inhabit soils that differ in their precipitation unpredictability, we used a permutation tests. We randomly shuffled host infection status (infected with *P. hayleyella* or infected with *P. bonniea*) from hosts infected by either of these species without replacement and calculated sample statistics for values of precipitation predictability across 10,000 samples. As sample statistics, we investigated the differences between both the means and medians of the two species after permutation. Both mean and median difference statistics gave equivalent results. We report the median difference *p*-value in the main text.

## RESULTS

To test for relationships between soil characteristics and symbiont prevalence, we used presence-absence data of symbionts that were collected from 22 collection trips to 21 locations (Fig. 1A, Table 1) across the eastern United States (*Haselkorn et al., 2019*, *2021*). Because some coinfections are known to be more common than expected (*Haselkorn et al., 2021*), we first tested all screened hosts for non-random coinfections that may also vary with the soil environment using logistic regression. *Paraburkholderia* coinfections were not more common than expected (Fig. 1B) even when we tested for coinfections at individual locations using Fisher's exact tests (Table S1). Generally, *Paraburkholderia* and *Amoebophilus* coinfections are more common than expected with *P. hayleyella* and *Amoebophilus* coinfections being the most enriched (Fig. 1B). This extends prior findings that focused on a subset of locations (*Haselkorn et al., 2021*). *Amoebophilus* and Chlamydiae coinfections are less common than expected across our sampled sites.

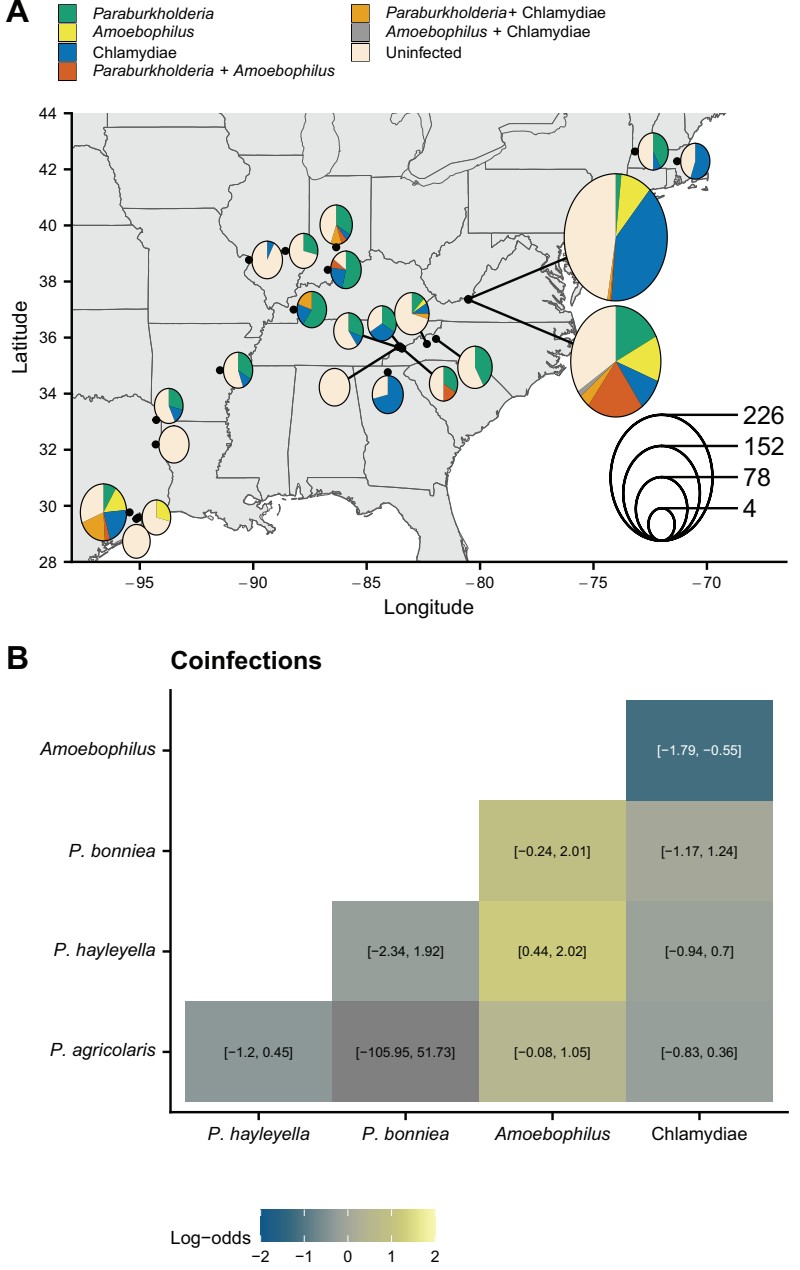

**Figure 1** ***D. discoideum* sample locations and patterns of endosymbiont infection.** (A) Map of *D. discoideum* sample locations. Black points show locations. Pie charts show the frequencies of symbionts in screened hosts. Relative pie chart size indicates the number of sampled hosts at a location. (B) Patterns of coinfection for different symbiont pairs from logistic regressions. Color shows the estimated log odds (95% confidence intervals are shown in the boxes). *P. agricolaris-P. bonniea* coinfection mean is not colored because it is an outlier due to the lack of any coinfections (confidence intervals are still shown). Map was generated with the sf package in R (*Pebesma, 2018*).

To identify associations with symbiont prevalence, we used logistic regression models. To measure precipitation unpredictability, we calculated Colwell's P (see Table 2 in *Colwell, 1974*) using monthly precipitation data for each location since 1901 (Fig. 2A).

**Table 1 Counts (and percent) of individual endosymbiont types, the total number of screened hosts, and the year of collection for each sampling location used in this study.**

| Location | P. hayleyella | P. agricolaris | P. bonniea | Amoebophilus | Chlamydiae | Total screened hosts | Year |
|---|---|---|---|---|---|---|---|
| Arkansas-Forest city | 3 (33.3%) | 0 (0%) | 0 (0%) | 0 (0%) | 1 (11.1%) | 9 | 2004 |
| Georgia-Cooper creek | 0 (0%) | 0 (0%) | 0 (0%) | 0 (0%) | 10 (71.4%) | 14 | 2000 |
| Illinois-Effingham | 0 (0%) | 2 (28.6%) | 0 (0%) | 0 (0%) | 0 (0%) | 7 | 2005 |
| Indiana-Bloomington (Lobelia) | 9 (50%) | 0 (0%) | 0 (0%) | 1 (5.6%) | 3 (16.7%) | 18 | 2005 |
| Indiana-Patoka lake | 8 (61.5%) | 4 (30.8%) | 0 (0%) | 1 (7.7%) | 3 (23.1%) | 13 | 2005 |
| Kentucky-Land between the lakes | 6 (60%) | 6 (60%) | 0 (0%) | 0 (0%) | 4 (40%) | 10 | 2004 |
| Massachusetts-Mt. Greylock | 1 (8.3%) | 4 (33.3%) | 0 (0%) | 0 (0%) | 1 (8.3%) | 12 | 2001 |
| Massachusetts-Boston | 0 (0%) | 0 (0%) | 0 (0%) | 0 (0%) | 5 (55.6%) | 9 | 2000 |
| Missouri-St. Louis | 0 (0%) | 0 (0%) | 0 (0%) | 0 (0%) | 1 (7.7%) | 13 | 2005 |
| North Carolina-Linville falls | 0 (0%) | 10 (41.7%) | 0 (0%) | 0 (0%) | 0 (0%) | 24 | 2001 |
| North Carolina-Little Butts gap | 0 (0%) | 1 (4.2%) | 3 (12.5%) | 1 (4.2%) | 3 (12.5%) | 24 | 2001 |
| Tennessee-Indian gap | 0 (0%) | 2 (20%) | 1 (10%) | 0 (0%) | 1 (10%) | 10 | 2001 |
| Tennessee-Rhodo thicket | 0 (0%) | 3 (42.9%) | 0 (0%) | 1 (14.3%) | 3 (42.9%) | 7 | 2001 |
| Tennessee-Road | 0 (0%) | 0 (0%) | 0 (0%) | 0 (0%) | 0 (0%) | 13 | 2001 |
| Tennessee-Sugarlands | 0 (0%) | 0 (0%) | 2 (33.3%) | 1 (16.7%) | 0 (0%) | 6 | 2001 |
| Texas-Armand Bayou | 0 (0%) | 0 (0%) | 0 (0%) | 2 (28.6%) | 0 (0%) | 7 | 2004 |
| Texas-Carthage | 0 (0%) | 0 (0%) | 0 (0%) | 0 (0%) | 0 (0%) | 12 | 2004 |
| Texas-Houston Arboretum | 2 (3.4%) | 19 (32.2%) | 0 (0%) | 14 (23.7%) | 27 (45.8%) | 59 | 2001 |
| Texas-Linden | 1 (14.3%) | 1 (14.3%) | 0 (0%) | 0 (0%) | 1 (14.3%) | 7 | 2005 |
| Texas-Webster | 0 (0%) | 0 (0%) | 0 (0%) | 0 (0%) | 0 (0%) | 4 | 2004 |
| Virginia-Mountain lake biological station | 0 (0%) | 3 (1.3%) | 4 (1.8%) | 23 (10.2%) | 92 (40.7%) | 226 | 2014 |
| Virginia-Mountain lake biological station | 26 (13.8%) | 48 (25.5%) | 8 (4.3%) | 70 (37.2%) | 33 (17.6%) | 188 | 2000 |

**Note:**
   Percents need not sum to 100 because of the presence of coinfections.

Rainfall was generally unpredictable as Colwell's P ranged from 0.25 to 0.42. To make sure that more recent unpredictability did not deviate from long-term predictability, we also calculated Colwell's P for the last 5 years before collection at each location (Fig. 2B). Colwell's P ranges from 0 to 1, with 0 being unpredictable and one being perfectly predictable (*Colwell, 1974*). Along with our measures of unpredictability, we collected mean annual precipitation (Fig. 2C), soil pH (Fig. 2D), soil mean annual temperature (Fig. 2E), soil carbon to nitrogen ratio data (Fig. 2F). Correlations between these variables tended to be low with the exception of mean annual temperature being correlated with pH and long-term Colwell's P (Fig. S1).

We found that the frequencies of the two *Paraburkholderia* species with reduced genomes, *P. hayleyella* and *P. bonniea*, were associated with unpredictable precipitation, but in opposite directions (Figs. 3A and 3B). Other variables measuring mean soil characters were not associated with prevalence unless also included with unpredictable precipitation (Table SI). *P. hayleyella* prevalence was higher in more unpredictable environments (log-odds = −1.047, se = 0.545; Fig. 3A) while *P. bonniea* prevalence was higher in more predictable environments (log-odds = 1.181, se = 0.442; Fig. 3B). These
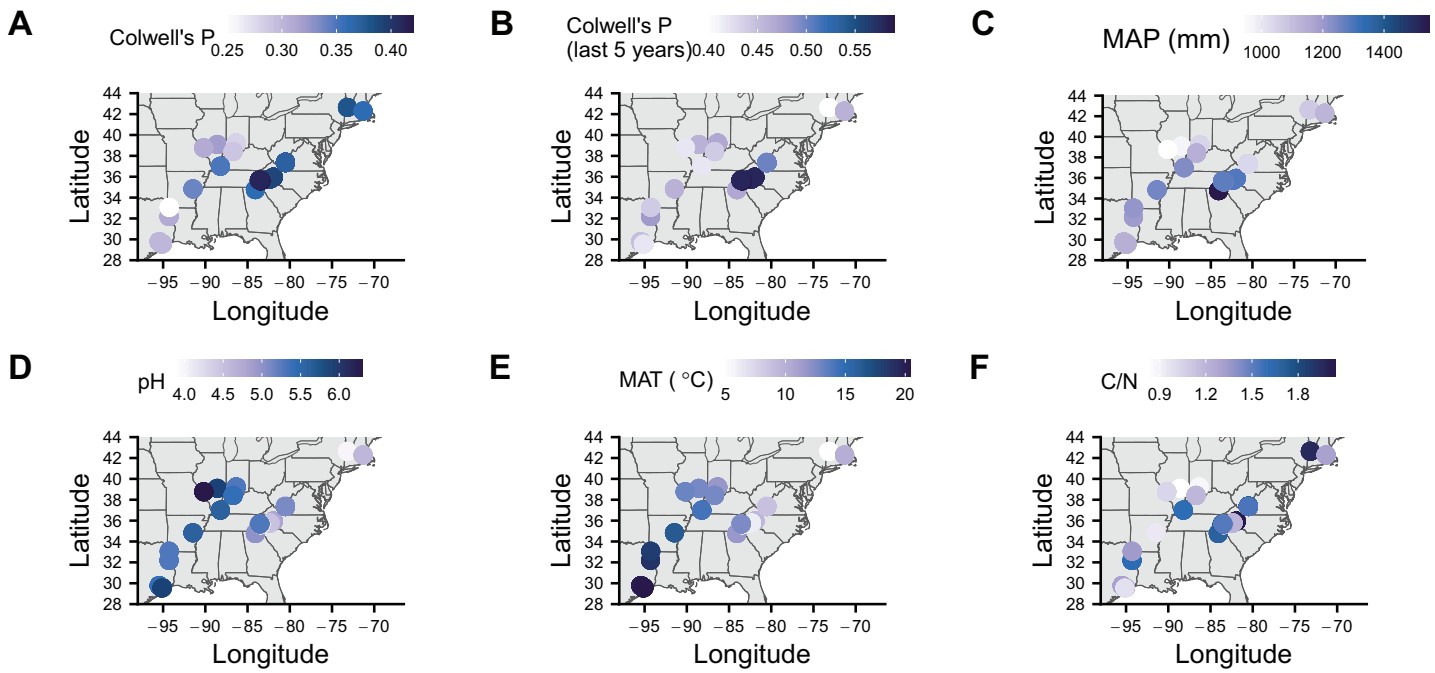

**Figure 2 Maps of soil characteristics from sample locations.** (A) Colwell's P for precipitation from 1901 to the year of sampling. (B) Colwell's P from the 5 years before sampling. (C) Mean annual precipitation (MAP) calculated from 1901 to the year of sampling. (D) pH of soil. (E) Mean annual temperature (MAT) of soil. (F) Carbon to nitrogen ratio (C/N) of soil. Map was generated with the sf package in R (*Pebesma, 2018*).

opposite responses to the predictability of precipitation remained even when we accounted for the influence of the two largest sampling locations (Fig. S2). Moreover, our data show (histograms in Fig. 3) that *P. hayleyella* is found where precipitation is relatively unpredictable (0.33.5 on average) while *P. bonniea* is found where precipitation is more predictable (38.1 on average; Permutation test: $p < 0.001$). For the other symbiont infections – *P. agricolaris*, the obligate *Amoebophilus* and Chlamydiae endosymbionts, *P. hayleyella-Amoebophilus* coinfections, and even symbiosis overall–prevalence was not associated with unpredictable precipitation or mean soil characteristics (Table SI).

## DISCUSSION

Our finding that *P. hayleyella* prevalence increases in unpredictable conditions supports our hypothesis that symbiosis may buffer hosts during times when conditions are unpredictable. However, our finding for the relatively less common *P. bonniea* is in the opposite direction of this hypothesis.

One explanation for why unpredictability differently affects *P. hayleyella* and *bonniea* prevalence is that these sister species (*Brock et al., 2020*) compete and are partitioning their niches within hosts based on unpredictable precipitation. Indeed, *P. hayleyella* and *P. bonniae* were found on different ends of our measure of the predictability of precipitation. Some additional support for niche partitioning comes from a previous finding that *P. hayleyella* and *P. bonniea* differ on which sugars they can metabolize (*Brock et al., 2020*). One argument against niche partitioning within hosts is that the prevalence of

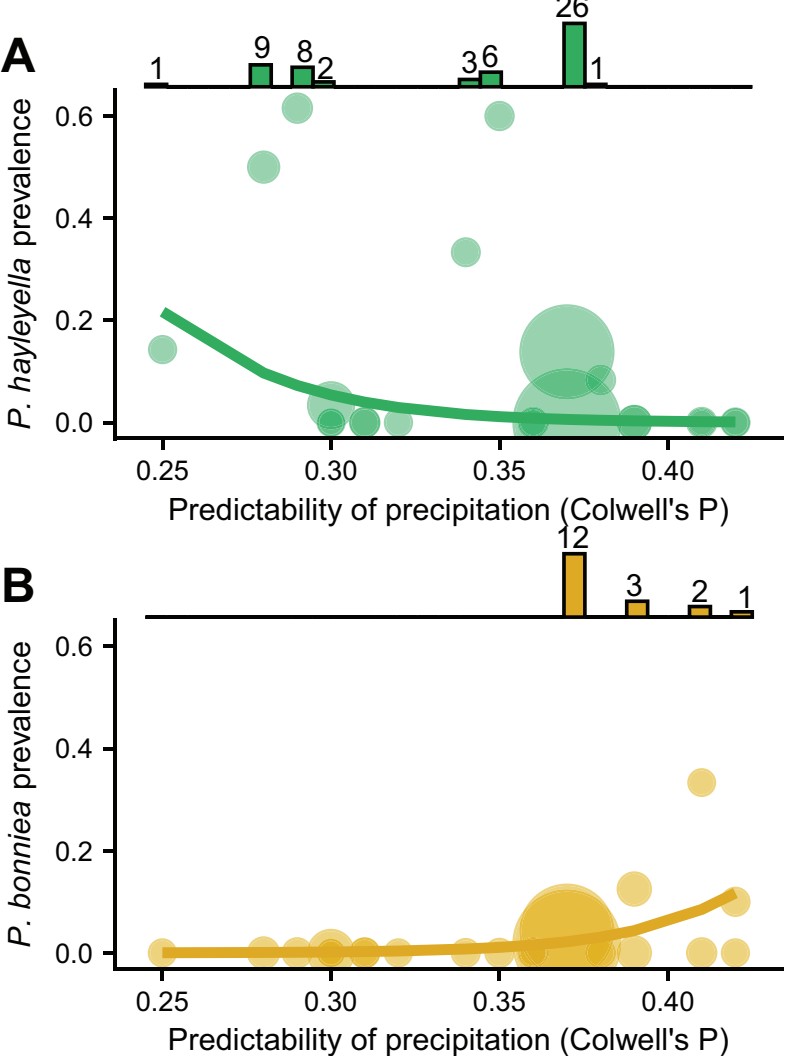

**Figure 3 _P. hayleyella_ and _P. bonniea_ are differently affected by and inhabit different areas of precipitation predictability.** The prevalence of _P. hayleyella_ (A) and _P. bonniea_ (B) with different values of predictability of precipitation across locations (samples per site is indicated by the size of the point). Prevalence is the fraction of screened hosts that were found with a given endosymbiont. Logistic regression fits are shown as lines. _P. hayleyella_ and _P. bonniea_ inhabit different soils in terms of their precipitation predictability as shown by histograms on top of panels (note that _P. bonniea_ is only found at predictability values above 0.37 while _P. hayleyella_ is found almost exclusively at or below this value). Numbers above bars in histogram are the number of symbionts found in screened hosts for a given value of precipitation predictability.

these symbionts may not be high enough for strong competition within hosts. Instead, competition in the soil could drive niche partitioning between these symbionts. In this case, interactions with other members of the _D. discoideum_ microbiome that were not included here may be involved. Another possibility is that life history characteristics of _P. hayleyella_ and _P. bonniea_ affect the ability of these bacteria to survive in soils with different levels of unpredictable precipitation.

In addition to the role of unpredictability, we also identified new associations between different symbionts in *D. discoideum* hosts. We found that *P. hayleyella* and *Amoebophilus* coinfections are more common than expected and *Amoebophilus* and Chlamydiae coinfections are less common than expected (Fig. 1B). The association between *P. hayleyella* and *Amoebophilus* suggests that the abundance of both may be driven by the same environmental conditions. Another possibility is that *P. hayleyella* and *Amoebophilus* are mutualists that have increased survival when together in the same host. The rarity of *Amoebophilus* and Chlamydiae coinfections may indicate competitive exclusion inside *D. discoideum* hosts. Another explanation is that *Amoebophlius* or Chlamydiae actively prevent each other's colonization. Chlamydial endosymbionts have been shown to reduce the success of other endosymbionts in other species of amoebae (König et al., 2019; Arthofer et al., 2022).

Our results provide suggestive evidence of the role of unpredictability driving symbiosis that should be followed up in future studies. However, our study is limited in several ways due to data constraints. First, our samples were not replicated over time, so our results do not capture the variation in *Paraburkholderia* symbiosis over time. This may be an important factor as other soil sampling studies have found that symbiosis with *Paraburkholderia* may vary over time in some locations (DuBose et al., 2022). Second, our soil and climate measures do not capture within site heterogeneity that is important in many microbial systems (Nannipieri et al., 2019). For this reason, our study should inspire future fieldwork with better sampling to better understand the drivers of symbiosis in this system.

This study demonstrates that the frequency of a microbial symbiosis can be associated with unpredictable environmental conditions. Unpredictable conditions may be an important driver of cooperation between members of the same species and between different species.

## ACKNOWLEDGEMENTS

We thank the members of the Strassmann-Queller lab along with Carlos Botero, Fred Inglis, and Jonathan Losos for feedback on this project.

### Funding

This material is based upon work supported by the National Science Foundation under grant numbers IOS 1656756, DEB 1753743, and DEB 2237266. Trey J. Scott is supported by the Mind Brain Behavior Interfaculty Initiative at Harvard University. The funders had no role in study design, data collection and analysis, decision to publish, or preparation of the manuscript.

## Grant Disclosures

The following grant information was disclosed by the authors:
National Science Foundation: IOS 1656756, DEB 1753743, and DEB 2237266.
Mind Brain Behavior Interfaculty Initiative at Harvard University.

## Competing Interests

The authors declare that they have no competing interests.

## Author Contributions

- Trey J. Scott conceived and designed the experiments, performed the experiments, analyzed the data, prepared figures and/or tables, authored or reviewed drafts of the article, and approved the final draft.
- Calum J. Stephenson conceived and designed the experiments, performed the experiments, analyzed the data, prepared figures and/or tables, authored or reviewed drafts of the article, and approved the final draft.
- Sandeep Rao performed the experiments, prepared figures and/or tables, and approved the final draft.
- David C. Queller conceived and designed the experiments, authored or reviewed drafts of the article, and approved the final draft.
- Joan E. Strassmann conceived and designed the experiments, authored or reviewed drafts of the article, and approved the final draft.

## Data Availability

The data and code are available at GitLab and Zenodo:
- https://Gitlab.com/treyjscott/symbiont_prevalence
- Scott, T. (2024). Data and code: Unpredictable soil conditions can affect the prevalence of a microbial symbiosis [Data set]. Zenodo. https://doi.org/10.5281/zenodo.11109505

## Supplemental Information

Supplemental information for this article can be found online at http://dx.doi.org/10.7717/peerj.17445#supplemental-information.

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
