# Peer review of "Unpredictable soil conditions can affect the prevalence of a microbial symbiosis"

_PeerJ, doi:10.7717/peerj.17445_

## Round 0.1 · original submission · Major Revisions

Dear Dr. Scott and colleagues:

Thanks for submitting your manuscript to PeerJ. I have now received three independent reviews of your work, and as you will see, the reviewers raised some relatively minor concerns about the research. This is great and indicates optimism for your work and the potential impact it will have on research studying soil microbiome dynamics.

While the concerns of the reviewers are relatively minor, this is a major revision to ensure that the original reviewers have a chance to evaluate your responses to their concerns. There are many suggestions, which I am sure will greatly improve your manuscript once addressed.

Therefore, I am recommending that you revise your manuscript, accordingly, taking into account all of the issues raised by the reviewers. I do believe that your manuscript will be greatly improved once these issues are addressed.

Good luck with your revision,

-joe

·

Basic reporting

General comments:

The authors describe a novel analysis of an existing data set of infection of Dictyostelium discoideum fruiting bodies with bacterial symbionts. The authors generate a new data set consisting of several characteristics of the soil at the locations and times where and when the fruiting bodies were sampled, and they analyze the two data sets for correlations. They find two examples of symbiotic species presence correlating with unpredictability of rainfall. I appreciate how clearly the authors delimit positive and negative results within the study. The overall motivation, context, and conclusions are very well described, and the whole study is well-written and presented.

Following, I have some generally minor suggestions to improve the clarity of the manuscript:

1. I was unsure which levels of the symbiosis variable were tested in the models. Did the authors test for correlation of the environmental factors with presence/absence of each taxon individually (the 3 Paraburkholderia species, Amoebophilus, and Chlamydiae), and also with each possible pair?
2. “Chlamydiae” is sometimes in italics and sometimes not (e.g. in abstract, line 75 vs lines 108 and 113). Is this because in some cases the authors refer to the phylum and in other cases to a genus of the same name within the phylum? The taxon being referred to is unclear.
3. Lines 36-37: three things are listed as not associated with any of the measures, but what they are and how they are different from each other is unclear. Consider rephrasing.
4. Line 63: “different species of bacteria have been identified” within fruiting bodies – it would be nice to know more about this, if possible. Where are the bacterial cells within fruiting bodies? When and how do they get transported by spores? How much of a fruiting body is made up of bacterial cells (abundance, diversity)? When was this symbiosis first discovered? I know these questions may not be answerable, in which case referring to them (or similar) either here or in the discussion would emphasize the ecological interest and relevance of studying this system and the contribution of the present work.
5. Line 74: small suggestion to state here (or clarify earlier in the text) that the obligate endosymbionts are also bacteria.
6. Line 79 says that Amoebophilus and Chlamydiae don’t “measurably affect host fitness” – does Paraburkholderia? Do spores being able to carry bacterial cells and “reduced spore production when edible bacteria are common” represent fitness metrics?
7. Line 83: “more common than expected” due to chance?
8. Lines 89-91: nice and clear statement of hypothesis and reasoning
9. Lines 95-97: about the soil nutrients – you might consider stating here that you will consider the C/N ratio as the measure of nutrients, and perhaps what that means about the soil (e.g. are some soils considered richer or poorer in nutrients, are there any other general trends of soil quality that might be relevant?).
10. Line 105: “21 locations” in the United States – suggest adding this because you later (lines 110-111) state why you exclude some locations from other countries.
11. Line 112: “clone”
12. Lines 127-130: why did you do the two measures of predictability (long-term and short-term)? Suggest clarifying this here or in the results section.
13. Line 134: “temperature data were generated” – by Lembrechts et al (2022) or by you? Suggest clarifying this. Nice justification in lines 136-137 for why you didn’t use the 5-15 cm data.
14. Line 153: suggest “characteristics” instead of “characters” for consistency.
15. Lines 155-158: do you know if any of these soil metrics are correlated with each other? E.g. does pH correlate with C/N ratio?
16. Line 191: you could state that Colwell’s P ranged from 0.25 to 0.45-ish, showing that rainfall was generally on the less-predictable side of things.
17. Lines 198-199: this felt out of place – suggest moving to the introduction.
18. Lines 204-205: this was already stated in line 202. Suggest revising.
19. Line 207: could add that the “more predictable environments” are from 0.35 and up.
20. Line 227: is it known how prevalent P. hayleyella and P. bonniea are in soils?
21. Lines 252-255: do you think it’s the unpredictability per se that is driving the correlation? Or that it’s unpredictable rainfall? Could experimental work test unpredictability of a range of environmental factors? If so it might be nice to suggest here, but not necessary.
22. Figure 1A: the cream color representing “uninfected” is hard to see. I suspect that this is intentional, to draw attention to the different infected types, but it makes the pie charts challenging to distinguish. I suggest adding a black border around the perimeter of each pie chart to help with this.
23. Figure 1A and B: suggest putting genus and species names in italics.

Experimental design

The research question and motivation are well-explained, as is the data acquisition and analysis methodology.

Validity of the findings

Conclusions are clear and supported. Limitations of the study are noted. Data are accessible and legible.

Reviewer 2 ·

Basic reporting

In general, the background information provided was good. However, I would suggest that more detail is given in regard to the Paraburkholderia symbionts for those not familiar with the system. For example, please expand on the explanation of the seed-out of food bacteria (L69), specifically when these symbionts benefit the host and when they have a cost to the host, as I found this somewhat unclear. Also importantly, you mention the Paraburkholderia are facultative, but not whether this means there is always a free-living stage or whether for the most part vertical transmission occurs.

Experimental design

I find it unclear which year the other environmental conditions (T, Ph, C/N etc) are from? Was the data from the sample year used? Please clarify

Validity of the findings

From figure 1, it appears that Cowell P values and mean annual precipitation is associated (e.g. it seems that higher precipitation is more predictable). Do the authors feel this should be accounted given that a strong correlation would go against the independence of these variables? Similarly, when testing the Cowells P last 5 years vs Colwells from 1901 in the same model does this not also raise an issue regarding independence of variables?

From Figure 3, can the authors comment on the fact that in the histograms above 3A it appears that the prevalence is perhaps bimodal. Furthermore, in both 3A and B the numbers of the endosymbionts seem small, and thus the effect of the larger sampling sites appears significant. Could the authors comment on the potential over-influence of the larger sampling sites given the small total number and whether the number of endosymbionts tested was a concern.

I commend the authors on their provision of the statistical tables in the supplementary data table and code, which has been done thoroughly. The only thing I find unclear is which model was used for Figure 1B comparisons. It is stated that Fisher exact test was used to test the Paraburkholderia coinfections, but what about the coinfection tests with Amoebophilis and Chlamydiae? Please make this clear.

I think that given the mixed results of the role of unpredictability the title should be toned down, perhaps to “can affect”

Additional comments

1.) Most importantly, I believe the figures could be improved to increase clarity:
a) Figure 1 A, I find this figure to be unclear. The pie charts make relative prevalence very difficult to ascertain, especially as the two blues used are difficult to distinguish. I would heavily recommend changing this figure type – perhaps with stacked bar graphs it would be easier to compare? And perhaps separating the map and prevalence graphs to be side panels?
If the authors keep the pie charts, then please at least ensure that they do not overlap like the two larger ones do, which prevents one from seeing some of the lower segments, and add outlines around them so they are more distinct compared to the background of the map. Furthermore, please use a more distinctive colour palette
b) Relating to the point above, given that the results focus heavily on P. hayleyella vs P. bonniea it would be very helpful to see their specific prevalence. I would recommend that the 3 species of Paraburkholderia are shown separately. This will make the pie charts more difficult to read, so again I suggest a different format.
c) Figure 1B, the authors provide confidence intervals but additional significance stars would also be helpful. The P. agricolaris – Chlamydiae confidence intervals values should be in black to improve readability.
d) Figure 2, this figure is clear. My only suggestion is to simplify the axis by removing the ‘N’ and ‘W’ and putting these in the axis titles.
e) Figure 3, the histograms have no units on the y axis, it would be helpful to know what magnitude of numbers are. In the legend I find it somewhat confusing the description of the histogram and then the permutation test.

2) It would be very helpful if a summary table could be provided. E.g. total number & percentages for each endosymbiont across all locations. This would help the assessment of sample size validity, and would provide support for the statement that refer to P. bonniea as “relatively less common” (L216).

3) Is there data available that has sampled the free-living endosymbionts within the same locations? It would an informative comparison to have the prevalence in the soil versus in the host, if this is possible.

4) Do the three species of Paraburkholderia vary in either the benefit/cost to the host? Perhaps the relative cost:benefit ratio of the symbionts is related to their differing relationship to precipitation predictability.


Minor points:

5) Please provide the sampling years for the symbiosis sampling (we are given the references, but the spread of time is relevant and at least the range should be mentioned in the text).

L83, add ‘by chance’ after more common than expected
L222, I think you mean to refer to Figure 3 and not 2 here

Reviewer 3 ·

Basic reporting

- Overall the language use is good and clear. The introduction is comprehensive and provides enough background and literature to understand the rest of the main text. Results of statical tests are shared but not the raw data.
- Line 76: I’d argue it’d be more prudent to have “have not yet had any measurable”, it can’t be ruled out from the previous work that you failed measuring an aspect of fitness that is more evident in the wild.
- I was confused about the repeated use of the term obligate symbiont for some species. This term is used in reference to the symbiont and/or host that cannot live independently of the relationship. Firstly, it wasn’t made totally clear that hosts can live independently (which I assume from the evidence presented here and in cited references). I’m also not sure if the authors have fully established if all the Amoebophilus and Chlamydiae strains observed are all definitely obligate as beyond some genomic signatures of a strong symbiotic association (e.g. Haselkorn et al 2019 & 2021) there is no mention if all the bacteria stains can survive outside of the host environment or not. Might they be confusing obligate with specialist in some cases? Some clarification and justification would be helpful, maybe around line 78. This is important as true obligate symbiosis (when a symbiont is trapped in a relationship with a host) have some important selective differences to partially/or preferentially symbiotic relationships, as any occasional horizontal transfer can have important evolutionary consequences (see Bennet&Moran 2015 Heritable symbiosis: The advantages and perils of an evolutionary rabbit hole)
- Figure2: Some presentation of the relationship between site environmental measurements would be valuable, for example multivariate plot.
- Figure3: Authors present the data as “prevalence”, would this be the proportion of samples infected per site? Please clarify. I’d like to also suggest having stacked bars in the histograms showing of total “number of samples”/ “number of symbionts”.

Experimental design

- The authors seem to be be using data gathered for other purposes, which has some down sides (i.e. lack of balanced sampling) though doesn’t discount it’s use. The main issue though is that a huge proportion of the data is from one site (over two time points), which equates to roughly 60% of the samples. Authors have called out the importance of this site. However, it would be worth, at least in SI, having the analysis without the Virginia-MLBS site to see how this site skews the results.
- Line 119: the source of the “online database” is not given, please provide. Also any details on how this data is collected would be useful (e.g. made how often are surveys of this data made and by whom?).

Validity of the findings

- Line 225/226: “One argument against niche partitioning is that the prevalence of these symbionts may not be high enough for strong competition over hosts”. Should this be “competition within” host. I’d also argue niche partitioning could occur more when competition is higher as one symbiont would be more likely pushed into a realised niche or specialise to be more competitive. The ideas here seem undeveloped/unsubstantiated I’d consider not making this argument at al. However, did the authors consider the role of life history in the observed difference in response to unpredictability? By this I mean one species might have a more sensitive diapause mechanism or other bet hedging stratergy (e.g. energy storage) that would incur a fitness cost in the short term but have a better ability to weather unpredictable precipitation events. Exploration of more literature around this area might be useful.

Additional comments

In the manuscript “Unpredictable soil conditions affect the prevalence of a microbial symbiosis” the authors study patterns of symbiosis in relation to climatic features and find interesting difference between genera. I consider this paper well-written and interesting, however, I suggest some clarifications to be made (as outlined above) before it is accepted for publication.

---

## Round 0.2 · accepted · Accept

Dear Dr. Scott and colleagues:

Thanks for revising your manuscript based on the concerns raised by the reviewers. I now believe that your manuscript is suitable for publication. Congratulations! I look forward to seeing this work in print, and I anticipate it being an important resource for groups studying soil microbiome dynamics. Thanks again for choosing PeerJ to publish such important work.

Best,

-joe

·

Basic reporting

The authors have addressed all of my suggestions in their revision.

Experimental design

No comment

Validity of the findings

No comment

Reviewer 2 ·

Basic reporting

I thank the authors for their thorough response to all of my comments. It particular, I appreciate that they tested the effect of the two larger sampling sites and also tested the potential influence of correlations, and neither of these factors was found to influence the conclusion. I think the addition of the table is useful for providing all the summary data, and I think the modifications to the figures have helped with their clarity. Finally, their clarifications to the text have answered any questions that I have, and I believe the manuscript is ready for publication.

My only minor comment is re. Figure S1, as I could not see/find its figure legend.

Experimental design

no comment

Validity of the findings

no comment

Reviewer 3 ·

Basic reporting

I am happy with the responses to my comments regarding this section, the new plot added to the supplementary is useful.

Experimental design

I am happy with the responses to my comments regarding this section. The random sampling approach is a nice approach to understanding the effect of the larger sample sites.

Validity of the findings

I am happy with the responses to my comments regarding this section and consider the result of interest and interpretation sound.

Additional comments

I find the responses to my comments acceptable and the paper is of value. Thank you for letting me read this paper.